# Availability of Financial and Medical Resources for Screening Providers and Its Impact on Cancer Screening Uptake and Intervention Programs

**DOI:** 10.3390/ijerph191811477

**Published:** 2022-09-12

**Authors:** Koshi Takahashi, Sho Nakamura, Kaname Watanabe, Masahiko Sakaguchi, Hiroto Narimatsu

**Affiliations:** 1Cancer Prevention and Control Division, Kanagawa Cancer Center Research Institute, 2-3-2 Nakao, Asahi-ku, Yokohama 241-8515, Japan; 2Kaneyama Town Clinic, 324-1 Kaneyama, Kaneyama Town, Mogami-gun 999-5402, Japan; 3Graduate School of Health Innovation, Kanagawa University of Human Services, 3-25-10 Tonomachi, Kawasaki-ku, Kawasaki 210-0821, Japan; 4Department of Genetic Medicine, Kanagawa Cancer Center, 2-3-2 Nakao, Asahi-ku, Yokohama 241-8515, Japan; 5Department of Engineering Informatics, Faculty of Information and Communication Engineering, Osaka Electro-Communication University, 18-8 Hatsucho, Neyagawa-shi 572-8530, Japan

**Keywords:** cancer screening, screening interventions, structural equation modeling, health services administration, population health management, preventive health services

## Abstract

Interventions for residents and medical/financial resources available to screening providers can improve cancer screening rates. Yet the mechanisms by which the interactions of these factors affect the screening rates remain unknown. This study employed structural equation modeling to analyze the mechanisms underlying these factors. Data for Japanese municipalities’ medical/financial status, their implementation of screening interventions, and the number of municipality-based cancer screening appointments from April 2016 to March 2017 were obtained from an open database. Five cancer screenings were included: gastric, lung, colorectal, breast, and cervical cancer screening; all are nationally recommended for population screening in Japan. We defined two latent variables, namely, intervention for residents and medical/financial resources, and then analyzed the relationships between these variables and screening rates using structural equation modeling. Models were constructed for gastric, lung, and breast cancer screening, and similar relationships were observed. With these cancer types, medical/financial resources affected the intervention for residents, directly affecting screening rates. One limitation of this study is that it only included screening by municipalities, which may cause selection bias. In conclusion, financial pressures and lack of medical resources may cause a reduction in screening intervention programs, leading to stagnant screening rates. Ensuring consistent implementation of interventions for residents may improve local and regional cancer screening rates.

## 1. Introduction

Evidence shows that cancer screening reduces mortality for several types of cancer through early detection and treatment [1]. The importance of cancer screening is increasing, given that the burden of cancer is expected to grow due to aging [2], since aging is one of the main risks of cancer.

Previous evidence indicates that several factors associated with participants and some interventions by screening providers affect the screening rate. For example, higher income is associated with higher cancer screening rates, specifically, the screening rates for cervical and breast cancer in the lowest income quartile were 61.6% and 53.8%, respectively, and in the highest income quartile they were 73.4% and 68.3%, respectively [3]. Older age also positively affects screening rates: “men and women 65 years and older had higher rates of any recommended colorectal cancer test (55.8% and 48.5%, respectively) than persons 50 to 64 years (males, 41.0%; females, 31.4%)” [4]. Further, high educational background [4,5] and high socioeconomic status [6,7] also positively affect screening rates. Interventions by screening providers, screening invitations and reminders for residents [8,9,10,11], co-payment strategies for cancer screening at the public expense [12], and education for the target population [8,13,14] positively affect screening rates. Additionally, screening providers’ financial status and the availability of medical resources also contribute to the screening rates. Previous studies have shown that financial pressures on screening providers and insufficient numbers of public health nurses are associated with low screening rates [15,16].

In clinical settings, these factors would interact to influence screening rates. However, it is unknown which mechanisms work in this process. Clarifying these mechanisms would make it possible to identify reasons for low screening rates and suggest efficient measures to improve them. Thus, describing these mechanisms is needed, particularly in Japan, where nationally organized screening programs are not yet in place [17]. In Japan, insurers (municipalities and companies) responsible for managing cancer screening are not obligated to implement such screening interventions for the insured or have them screened. Therefore, insurers can decide whether to implement screening interventions at their discretion [17]. This situation allows screening providers to make insufficient efforts to improve screening rates in order to avoid the financial or medical burden of providing screenings. Thus, there may be an intervening factor in the previously reported relationship between financial/medical resources and screening rates, such as screening interventions [15,16]. Analyzing these mechanisms may help correctly recognize the problems related to an ineffective screening system, and help to develop efficient strategies to improve the screening rates. For example, if sequential mechanisms exist, such as medical/financial resources affecting the screening intervention and subsequently having an effect on screening rates, providing resources can help to both increase interventions and improve the screening rates. Alternatively, if each of these factors affects the screening rates independently, it would be necessary to mandate the screening interventions in addition to providing support for resources.

In this study, we used structural equation modeling (SEM) to analyze how Japanese municipalities’ medical/financial resources and screening intervention policies affect cancer screening rates. Our objective was to elucidate the causal relationships between cancer screening rates and multiple factors of the municipalities policies and conduct. SEM is a statistical technique used to model hypothesized relationships among observed and unobserved variables. Variables that cannot be observed are treated as latent variables in SEM, and constructed from measured variables. The accuracy of SEM results is evaluated based on fit indices and overall fit of the model, which leads to a valid analysis of the relationships. Given these characteristics, SEM was suitable for this study, which aimed to assess causal relationships between screening rates and factors that can affect them, including unmeasurable ones, such as the availability of medical/financial resources of municipalities or how municipalities provide screening interventions.

## 2. Materials and Methods

### 2.1. Study Design

An ecological study was conducted using an open database. Most of the data were collected from e-stat, an open online database provided by the Statistics Bureau of the Ministry of Internal Affairs and Communications [18]. The sources of all the acquired data are listed in the Appendix A.

### 2.2. Cancer Screening

In Japan, insurers, mainly local municipalities and corporate employers, implement population-based cancer screening. The data on participants in company-based cancer screening is not formally recorded; therefore, our analysis focused on the number of participants in municipality-based screening to calculate the screening rate in this study. As some data were exclusively available at the prefecture level, we combined data from each local municipality by prefecture, resulting in data on 47 prefectures. Five cancer screenings were included: gastric cancer, lung cancer, colorectal cancer, breast cancer, and cervical cancer; implementation of these testing programs is nationally recommended for population screening in Japan [17,19]. Since gender differences have been reported in previous studies [20,21,22], we analyzed data according to gender as well.

The number of participants screened was derived from the Report on Regional Public Health Services and Health Promotion Services conducted in 2016 [18]. In Japan, a two-day fecal occult blood test for over 40-year-olds is recommended for colorectal cancer screening. A biennial Pap smear test for people over the age of 20 is recommended for cervical cancer. The number of participants in these screenings was used to calculate the screening rates for colorectal and cervical cancer. Screening methods for gastric, lung, and breast cancer are not standardized in Japan; thus, we defined the number of people screened for these cancers based on their screening recommendations [23]. 

For breast cancer, the sum of the people who undertook biennial mammography plus-minus visual palpation was used. The number of people over 40 years taking an annual chest X-ray examination was used for lung cancer. Some municipalities conduct sputum cytology for heavy smokers, but the eligibility criteria for the test differ among municipalities. Therefore, in an effort to maintain uniformity of data and accuracy, we did not count these people. The number of people over 40 years old who underwent an annual gastric X-ray examination was used for gastric cancer. Due to issues in data availability, we did not include the number of participants in the endoscopic screening (Appendix A), which is recommended biennially for those over 50 years old. 

While all residents can participate in the municipality-based screening, it is often customary for employed individuals to take the screening provided by their insurer, which in most cases is their company. Those who do not have the opportunity to be screened by companies, such as those self-employed or unemployed, are the target of these municipality-based screenings. Thus, the total population minus the number of people employed will indicate the population eligible for municipality-based screening (1). Screening rates were calculated after being stratified by cancer type, prefecture, and sex, as follows (2):(1)Number of people eligible for cancer screening =Total population−Number of the employed+Number of the primary industry workers
(2)$$\text{Cancer}\;\text{screening}\;\text{rate}=\frac{\text{Number}\;\text{of}\;\text{people}\;\text{who}\;\text{received}\;\text{the}\;\text{cancer}\;\text{screenings}}{\text{Number}\;\text{of}\;\text{people}\;\text{eligible}\;\text{for}\;\text{the}\;\text{cancer}\;\text{screenings}}$$

The number of employed individuals and primary industry workers was obtained from the 2015 Census.

### 2.3. Indicators of Financial Resources

Three financial indicators were obtained from the 2016 Survey of Local Financial Conditions [18], and two variables to be used in the analysis were created by us using these indicators. The first variable was the sum of the municipal Health and Sanitation Expenditure and the Public Health Center Expenditure per capita (public health expenditures). Health and Sanitation Expenditure represents local municipalities’ expenditures for various health projects, and the costs of projects related to cancer screening are also included in this category. Public Health Center Expenditure is the operating expense of public health centers, which are responsible for public health projects in each municipality. In some municipalities, public health centers are responsible for cancer screening services, and in such cases, the costs concerning cancer screening are recorded as Public Health Center Expenditures. Since a fixed standard for the categorization of expenses of cancer screening costs is lacking, assigning expenditure categories is at each municipality’s discretion. For this study, these variables were summed and treated as a single variable. We assumed that public health expenditures reflect the cost invested in public health projects, including cancer screening. The second variable was the availability of municipalities’ general revenue divided by their population (general revenue per capita). The general revenue is the municipality’s budget and its use is decided at each municipality’s discretion. Cancer screening services in Japan are implemented using this budget in all municipalities. We assumed that the general revenue per capita reflects the financial capacity to implement the projects, including projects aimed at improving cancer screening.

### 2.4. Indicators of Medical Resources

Four variables were obtained from surveys conducted in 2016 as indicators of health care resources [18]. The number of nurses and public health nurses per 1000 individuals in each prefecture was noted. We included public health nurses, as previous studies show that they have an impact on the screening rate, as they are responsible for the practical work of cancer screenings, such as sending screening invitations to the residents. The third variable was the number of hospitals, including clinics, per 1000 individuals. We excluded hospitals and clinics specializing in psychiatry as they rarely provide cancer screening. The last variable for medical resources was the number of physicians per 1000 individuals in each prefecture.

### 2.5. Indicators of Screening Interventions

Eight indices were used as indicators of screening interventions. All indicators were obtained from the Survey on Cancer Screening Practices in Municipalities [24], a national survey. We used the rate of municipalities that met the following conditions: sending screening invitations (Call), re-invitations to the unscreened after the call (Recall), providing cancer screenings free of charge (Charge-free), providing screenings on an evening or holidays (After-hours), providing an opportunity for the residents to take screening in other municipalities (Extra-region), providing cancer screenings using a modality outside the recommendation of the national guideline (Modality extension), providing cancer screenings with little evidence of mortality reduction (Out of evidence), and limiting the number of people who can participate in the cancer screenings (Upper-limit set).

### 2.6. Other Indicators

The aging rate was obtained from the 2015 Census [18]. Average annual household income by prefecture was obtained from the 2016 National Household Income Structure Survey [18].

### 2.7. Statistical Analysis

The SEM was conducted to visualize the relationship between these variables and the screening rates. We defined two latent variables—medical/financial resources and screening interventions—and hypothesized that these latent variables affected screening rates directly, while medical/financial resources affected screening interventions. Other details of the prespecified model based on the hypothesis are shown in Figure 1.

To evaluate the goodness of fit of the data to the model, an χ^2^ test was conducted to examine the model’s reliability, and the model fit indices were calculated. The fit indices assessed in this study were the goodness-of-fit index (GFI), adjusted goodness-of-fit index (AGFI), standardized root mean square of residual (SRMR), comparative fit index (CFI), and root mean square error of approximation (RMSEA).

In the χ^2^ test, the cut-off of the *p*-value was >0.05. For SRMR and RMSEA, the cut-off was <0.08, indicating a good fit [25]; and ≥0.10, indicating a poor fit [26]. For GFI, AGFI, and CFI, we considered a cut-off value of >0.90, indicating a good model fit [27]. The GFI and AGFI are strongly affected by sample size, and it is suggested that these factors should be assessed with other fit indices [26]. Therefore, we decided to use the *p*-value of the χ^2^ test, RMSEA, SRMR, and CFI as measures to assess the acceptability of the analysis, with a cut-off of > 0.05, <0.10, <0.10, and >0.90, respectively. When all indices met the criteria, the fit of the data was considered to be good and acceptable. In the analysis, we assumed a correlation of the residual errors between the call and recall, the call and upper limit, and the general revenue and public health expense. The maximum likelihood estimation method is used in SEM, and this method requires that the data follow a multivariate normal distribution. However, some of the data used in this study, such as policy implementation rates and financial indicators, were not expected to follow a normal distribution. We performed normalization and standardization using the Box-Cox transformation to deal with such variables. For some of the variables related to screening interventions, the distributions were highly skewed and could not be approximated to a normal distribution by Box-Cox transformation. The variables that could not be transformed into normal distribution were “upper-limit” in lung and colorectal cancer screening; “extra-region,” “after-hours,” “out of evidence,” and “modality extension” in all types of cancer screenings; and “charge-free” for all cancers except lung cancer. Since these variables had substantial biases, it was considered difficult to analyze their effects on the outcomes, so we decided to exclude these variables from the analysis. Data were analyzed using R (Ver. 4.0.3) (R Core Team, Vienna, Austria) [28], lavaan package and semPlot package [29,30].

## 3. Results

### 3.1. Characteristics of Collected Data

Characteristics of the obtained data are shown in Table 1 and Table 2. The variables related to cancer screening interventions differed by prefectures and cancer types. Calls ranged from the lowest of 43.5% to the highest of 100%. For recall, it ranged from merely 5.6% to 93.3%. The upper limit ranged from 0% to 100% and was exceptionally high for breast cancer screening, with a median of 52%. It was notable that out of evidence showed a median of 90.5%, screening for cancers such as thyroid cancer, endometrial cancer, and prostate cancer.

### 3.2. Results of SEM

For gastric cancer, lung cancer, and breast cancer, the fit indices achieved acceptable levels, and the structure of the final model was similar for these cancer types. We could not construct a model that achieved sufficient fit indices for colorectal and cervical cancers (Figure 2, Figure 3, Figure 4, Figure 5 and Figure 6). Regarding the cancer types for which models could be constructed, interventions for the residents directly impacted the screening rate. Annual household income had a direct impact on the screening rate, except in the case of lung cancer screening among men. However, the impact was consistently smaller than that of the intervention for residents. Compared with annual household income, the standardized path coefficients of the intervention on the screening rate were approximately twice as large for gastric cancer screening, approximately 2.8 times the rates for lung cancer screening for women, and approximately 3.1 times the rates for breast cancer screening. Further, interventions for residents were influenced by medical/financial resources, and the aging rate influenced the medical/financial resources. The aging rate directly impacted the screening rate for lung cancer screening for men and breast cancer screening.

The indicators related to latent variables were similar among the cancer types. In every model, public health expenditure, number of public health nurses, and general revenue per capita were affected by medical/financial resources. The standardized path coefficients between these indicators and the latent variable ranged from 0.60 to 0.95 (*p*-value < 0.01). The latent variable indicating screening interventions affected “call”, “recall”, and “upper-limit” in gastric and breast cancer screening; it also affected “call”, “recall”, and “charge-free” in lung cancer screening. It positively affected “call,” “recall”, and “charge-free,” while negatively affecting upper-limit. The goodness of fit indices for constructed models are shown in Table 3.

## 4. Discussion

In this study, we visualized the relationships between cancer screening rates and multiple factors relating to screening providers. For gastric, lung, and breast cancer, the fit indices of the constructed model met acceptable levels, and similar relationships were observed regardless of cancer types. Medical/financial resources affected screening interventions in these cancer types, and interventions affected screening rates. According to these results, we can presume that the screening interventions such as call and recall are implemented depending on the medical/financial resources of the providers (municipalities), and the screening intervention mainly determines the screening rate.

Results suggested that the number of public health nurses comprises one of the medical resources influencing interventions in the population. Previous studies showed that the number of public health nurses affected the screening rates [15,16]. Our results are consistent with these previous reports, suggesting that the relationship between the number of public health nurses and the screening rates is probably mediated by the screening interventions. Public health nurses are engaged in tasks related to screening interventions. Therefore, municipalities facing a shortage of public health nurses may be unable to implement these interventions adequately. The number of medical nurses, a similar indicator, was excluded in the final models. However, it was consistently related to the medical/financial resources in the model building process, regardless of cancer types (Appendix B).

This study suggested that general revenue and public health expenditures are part of the financial resources that influence screening interventions. This result is consistent with previous studies indicating that financial pressure due to the lack of subsidies negatively influences screening rates [15,16]. The general revenue is the budget each municipality executes at its discretion. In Japan, cancer screening programs compete financially with other programs funded by the general revenue. It is claimed that this is one of the causes of Japan’s low cancer screening rate [15,16,31]. It is not easy for public health providers to obtain a screening budget out of their limited financial resources. Under these circumstances, the providers may reduce their screening interventions to lessen their short-term expenditures. A similar case has been observed in Greece, which recently suffered a serious financial collapse and implemented a policy limiting screening participation to reduce short-term health care costs. This policy has been criticized for its risk of increasing cancer cases and future health care costs [32]. These previous studies and cases support the hypothesis derived from this study that financial pressures on screening providers will reduce their efforts to improve the screening rates.

This study included the variables related to residents, household income, and age as factors affecting screening rates. Household income positively influenced screening rates for every cancer type, consistent with previous results [3]. The results for age are consistent with previous studies and provide further information. There was an indirect, positive effect through latent variables and a direct negative effect on screening rates, as observed in the model for breast cancer. The sum of these two effects remained positive for screening rates, consistent with previous results [4]. However, the results of this study indicate that age may affect screening rates through multiple mechanisms, suggesting that the effect of age on screening rates may change with the target population and screening system differences.

Our findings showed that there may be a sequential, causal relationship in the cancer screening program, starting from medical/financial resources, through intervention, and then screening rates. A survey supporting this result was conducted among Japanese municipalities that limited screening participation, with 64% reporting that the limitation was due to the limited capacity of the screening centers and 27% reporting that it was due to financial restrictions [33]. However, the mechanisms at work between these factors have not been previously analyzed. Although previous studies report that many factors affect screening rates, how these factors interact or the underlying mechanisms revealing how they affect the screening rate remained unknown. This study is the first to analyze multiple factors affecting screening rates not independently, but as a model that considers the complex mechanism, including the interacting effects. This result would not be achieved by conventional regression analysis, and the novelty of this study is the description of this mechanism by introducing SEM. These results enable us to undertake a more informed strategy to improve cancer screening. Our results indicate that expanding medical/financial resources may help implement screening interventions and consequently improve screening rates. Therefore, identifying and resolving the resource bottleneck of each municipality may help improve screening rates. For example, for those municipalities experiencing difficulty making invitations due to a shortage of public health nurses, recruiting more public health nurses may be an effective way to improve the screening rates. Regarding the system, financial subsidies limited to the use of screening may lead to an increase in screening rates. Previous findings support this presumption in that cutting cancer screening expenses by 10% in the municipalities surveyed was associated with a 9.3% decrease in screening attendance compared with the previous year [15]. This result is consistent with the presumption made throughout our study.

This study had some limitations. First, this study covered only screening by municipalities, and screening performed by companies was excluded, as the associated data for that screening were unavailable, potentially causing a selection bias. Second, this study did not consider some individual factors such as educational background [4,5] and socioeconomic status [6,7] that are known to affect screening rates. Our preliminary analysis using a multiple regression included educational backgrounds, but their impact on screening rates differed depending on the screening system, whether it was conducted by a municipality or a company. Thus, we excluded this variable from the analysis to reduce systemic error. The small sample size, given the number of prefectures in Japan, was also a limitation of our study that restricted the number of estimators in the SEM. We attempted to reduce the estimators to one-fifth of the observation (number of prefectures) to acquire the most reliable result possible [34]. The results we obtained were reasonable and consistent, and the fit indices were acceptable for gastric, lung, and breast cancer. Therefore, although the sample size was small, we consider it to be reasonable. Finally, there was a limitation in the review process. Other studies may have analyzed the relationship between the screening rates and the factors affecting it using different statistical methods. In addition, since we searched only English and Japanese literature, we did not examine previous studies in other languages. However, to the best of our knowledge, no previous studies have analyzed the entire relationship among factors affecting screening rates as a model, nor are there any studies that used SEM to investigate the relationship between screening rates and the factors affecting them.

## 5. Conclusions

Our findings indicate that interventions by screening providers to promote cancer screening affect screening rates and that interventions to improve cancer screening rates directly impact screening rates. In addition, the availability of health care resources and the economic status of screening providers may affect screening rates due to interventions. These results indicate that by improving the financial situation of screening providers and expanding their medical resources, screening providers may strengthen their interventions and thus, improve screening rates. The results of this study suggest that identifying and supporting the financial/medical resources lacking in each municipality could improve screening interventions and, consequently, increase screening rates.

## Figures and Tables

**Figure 1 ijerph-19-11477-f001:**
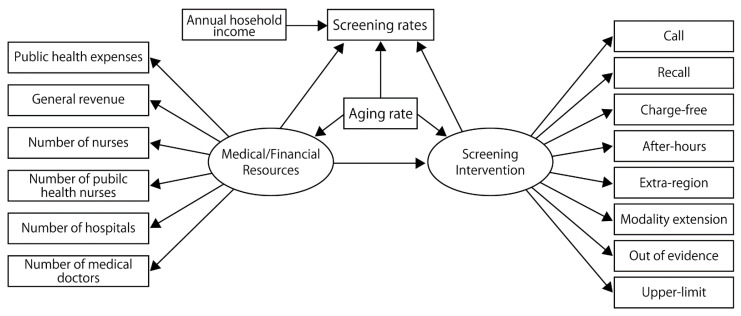
A pre-analysis model built on the initial hypothesis. Call, screening invitations to the residents; Recall, re-invitations to those unscreened after the call; Modality extension, providing cancer screening using a modality outside the recommendation of the national guideline; Out of evidence, providing screening with little evidence of mortality reduction; Upper limit set, limiting the number of people who can participate in the cancer screenings; Extra-region, providing an opportunity for the residents to take screening in other municipalities; After-hours, providing screenings on holidays or in the evenings; charge-free, providing cancer screenings free of charge.

**Figure 2 ijerph-19-11477-f002:**
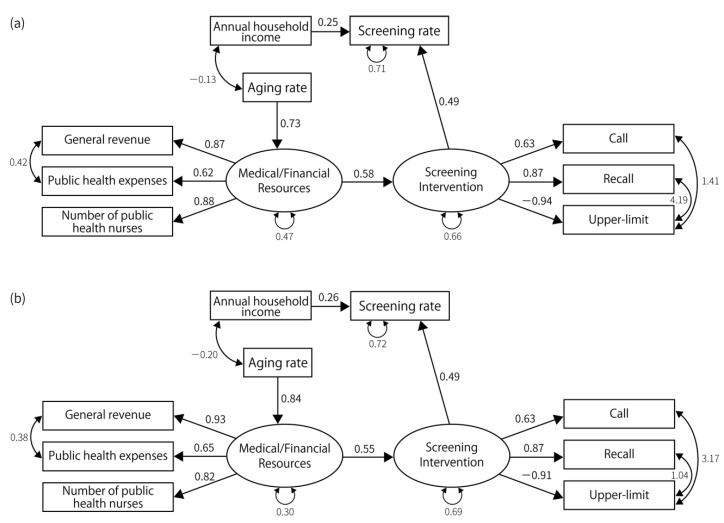
The final model for gastric cancer. (**a**) The final model for gastric cancer screening in males. GFI (goodness of fit index) = 0.872, AGFI (adjusted goodness of fit index) = 0.738, PGFI (parsimonious goodness of fit index) = 0.426, SRMR (standardized root mean square of residual) = 0.09, CFI (comparative fit index) = 0.96, RMSEA (root mean square error of approximation) = 0.080, *p*-value by χ^2^ test = 0.155, χ^2^ = 28.66, degrees of freedom = 22. (**b**) The final model for gastric cancer screening in females. GFI = 0.892, AGFI = 0.780, PGFI = 0.436, SRMR = 0.083, CFI = 0.96, RMSEA = 0.087, *p*-value by χ^2^ test = 0.123, χ^2^ = 29.82, degrees of freedom = 22.

**Figure 3 ijerph-19-11477-f003:**
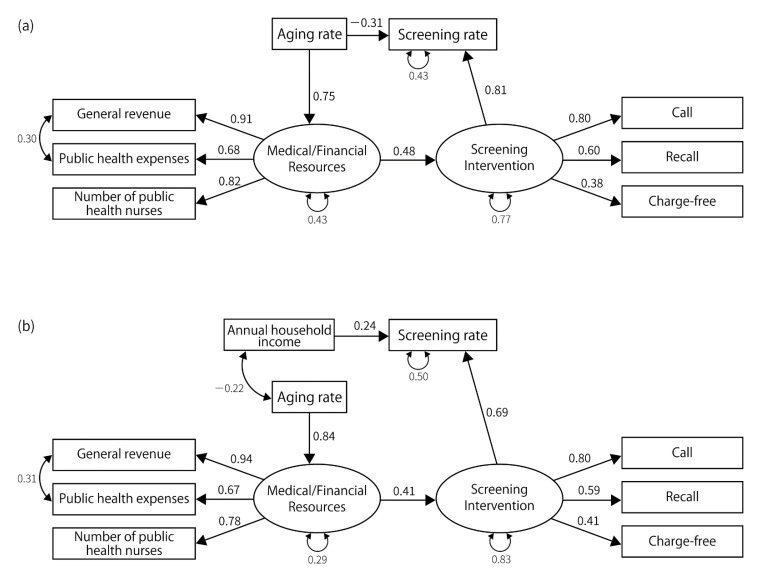
The final model for lung cancer. (**a**) The final model for lung cancer screening in males. GFI = 0.943, AGFI = 0.878, PGFI = 0.445, SRMR = 0.070, CFI = 1.00, RMSEA = 0.000, *p*-value by χ^2^ test = 0.683, χ^2^ = 13.77, degrees of freedom = 17. (**b**) The final model for lung cancer screening in females. GFI = 0.887, AGFI = 0.789, PGFI = 0.473, SRMR = 0.093, CFI = 0.94, RMSEA = 0.096, *p*-value by χ^2^ test = 0.077, χ^2^ = 34.438, degrees of freedom = 24.

**Figure 4 ijerph-19-11477-f004:**
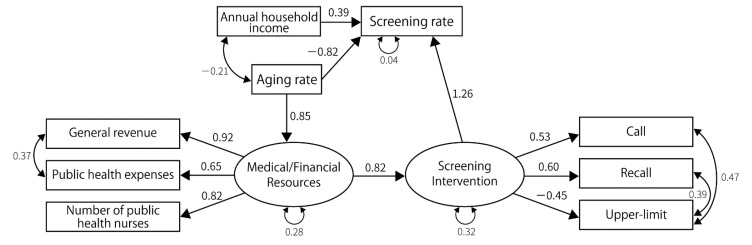
The final model for breast cancer. GFI = 0.930, AGFI = 0.850, PGFI = 0.434, SRMR = 0.080, CFI = 0.954, RMSEA = 0.096, *p*-value by χ^2^ test = 0.091, χ^2^ = 30.03, degrees of freedom = 21.

**Figure 5 ijerph-19-11477-f005:**
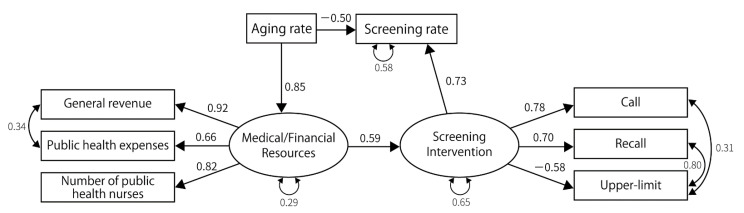
The final model for cervical cancer. GFI = 0.909, AGFI = 0.782, PGFI = 0.379, SRMR = 0.095, CFI = 0.896, RMSEA = 0.168, *p*-value by χ^2^ test = 0.003, χ^2^ = 34.798, degrees of freedom = 15.

**Figure 6 ijerph-19-11477-f006:**
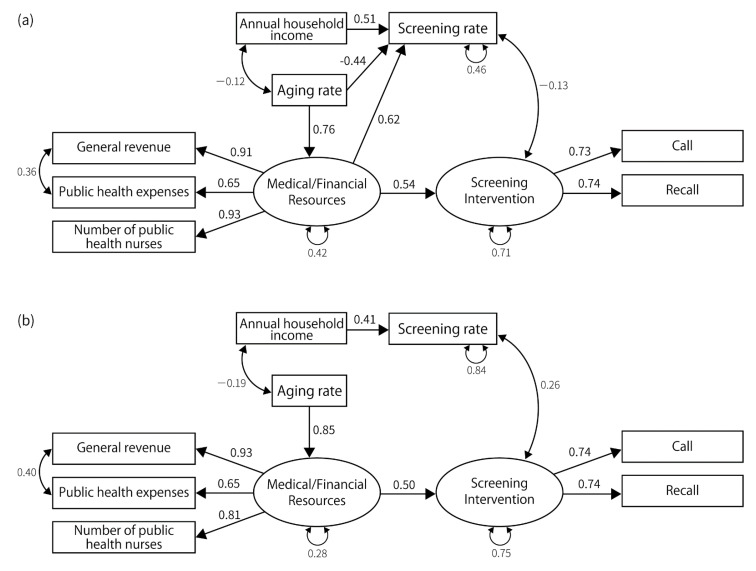
The final model for colorectal cancer. (**a**) The final model for colorectal cancer screening in males. GFI = 0.874, AGFI = 0.717, PGFI = 0.388, SRMR = 0.108, CFI = 0.911, RMSEA = 0.133, *p*-value by χ^2^ test = 0.022, χ^2^ = 29.362, degrees of freedom = 16. (**b**) The final model for colorectal cancer screening in females. GFI = 0.876, AGFI = 0.753, PGFI = 0.438, SRMR = 0.110, CFI = 0.898, RMSEA = 0.142, *p*-value by χ^2^ test = 0.009, χ^2^ = 35.111, degrees of freedom = 18.

**Table 1 ijerph-19-11477-t001:** Description of the screening rates and variables affecting screening interventions across 47 prefectures used in the analysis.

	Gastric Cancer		Lung Cancer		Colorectal Cancer		Breast Cancer	Cervical Cancer
	Men	Women	Men	Women	Men	Women		
Number of Prefectures	*n* = 47	*n* = 47	*n* = 47	*n* = 47	*n* = 47	*n* = 47	*n* = 47	*n* = 47
Cancer screening rate (%)	9.3 (6.8, 11.5)	7.4 (5.6, 9.4)	23.2 (13.5, 28.9)	21.2 (12.8, 27.0)	20.3 (17.6, 27.8)	21.2 (17.0, 26.6)	24.2 (20.6, 28.9)	26.3 (23.1, 30.3)
**Variables Related to Screening Intervention ***								
Call (%)	80.0 (71.6, 90.0)		79.7 (69.1, 90.0)		81.5 (74.2, 92.2)		81.5 (69.5, 88.4)	81.0 (72.7, 90.0)
Recall (%)	31.4 (22.1, 52.1)		28.0 (18.7, 44.9)		42.1 (28.2, 52.8)		44.0 (33.3, 55.1)	44.8 (34.5, 55.6)
Charge-free (%)	7.0 (3.0, 20.5)		28.0 (19.3, 45.9)		11.1 (5.1, 24.5)		6.3 (0.9, 16.5)	8.0 (3.2, 20.0)
>0% (n, %)	38 (80.9)		-		45 (95.7)		35 (74.5)	38 (80.9)
After-hours (%)	87.0 (80.0, 94.0)		86.4 (73.7, 92.1)		87.0 (77.6, 93.1)		81.8 (62.9, 90.5)	76.0 (57.4, 89.5)
>median (n, %)	24 (51)		24 (51)		24 (51)		24 (51)	24 (51)
Extra-region (%)	1.7 (0.0, 5.0)		1.7 (0.0, 5.0)		0.00 (0.0, 5.2)		3.1 (0.0, 7.3)	1.7 (0.0, 5.8)
>0% (n, %)	24 (51.1)		24 (51.1)		22 (46.8)		31 (66.0)	24 (51.1)
Modality extension (%)	15.1 (7.0, 27.8)		12.5 (4.5, 20.6)		1.7 (0.0, 7.4)		80.0 (50.1, 94.7)	11.1 (7.5, 23.8)
>0% (n, %)	42 (89.4)		40 (85.1)		24 (51.1)		44 (93.6)	39 (83.0)
Out of evidence (%)	90.5 (83.7, 100.0)		90.5 (83.7, 100.0)		90.5 (83.7, 100.0)		90.5 (83.7, 100.0)	90.5 (83.7, 100.0)
<100% (n, %)	13 (27.7)		13 (27.7)		13 (27.7)		13 (27.7)	13 (27.7)
Upper limit (%)	28.6 (13.5, 52.9)		13.3 (3.5, 28.8)		7.3 (2.0, 18.0)		52.0 (30.2, 67.7)	29.6 (12.2, 52.0)
>0% (number, %)	-		36 (76.6)		36 (76.6)		-	-

Data represented as median (interquartile range) unless otherwise specified. * Variables denoted as – are used as continuous variable and thus were not categorized. Call, sending screening invitations; Recall, re-invitations to the unscreened after the call; Charge-free, providing cancer screenings free of charge; After-hours, providing screenings on an evening or holidays; Extra-region, providing an opportunity for the residents to take screening in other municipalities; Modality extension, providing cancer screening using a modality outside the recommendation of the national guideline; Out of evidence, providing cancer screenings with little evidence of mortality reduction; Upper-limit; limiting the number of people who can participate in the cancer screenings.

**Table 2 ijerph-19-11477-t002:** Data on availability of resources across 47 prefectures used in the analysis.

Data Type	Values
**Basic Statistics**	
Mean annual household income (10^3^ yen)	5.2 (4.9, 5.7)
Population aged ≥ 65 (%)	25.5 (24.3, 26.6)
**Medical/financial resources**	
Public health expenses (10^3^ yen)	20.6 (17.5, 22.9)
General revenue per capita (10^3^ yen)	246.9 (218.3, 266.7)
Number of nurses (per 10^3^ people)	10.3 (8.6, 11.6)
Number of public health nurses (per 10^3^ people)	0.5 (0.4, 0.6)
Number of hospitals and clinics (per 10^3^ people)	0.9 (0.8, 1.0)
Number of medical doctors (per 10^3^ people)	2.4 (2.2, 2.8)

Data represented as median (interquartile range).

**Table 3 ijerph-19-11477-t003:** Results of SEM.

Latent Factors	Measured Variables	Gastric Cancer				Lung Cancer				Colorectal Cancer				Breast Cancer		Cervical Cancer	
		Men		Women		Men		Women		Men		Women					
		Estimate	*p*-Value	Estimate	*p*-Value	Estimate	*p*-Value	Estimate	*p*-Value	Estimate	*p*-Value	Estimate	*p*-Value	Estimate	*p*-Value	Estimate	*p*-Value
**Latent variable measurement model**																	
Resource	Number of public health nurses	0.88	-	0.79	-	0.82	-	0.78	-	0.80	-	0.79	-	0.80	-	0.83	-
	Public health expenses	0.62	<0.001	0.72	<0.001	0.68	<0.001	0.67	<0.001	0.73	<0.001	0.72	<0.001	0.72	<0.001	0.67	<0.001
	General revenue per capita	0.87	<0.001	0.96	<0.001	0.91	<0.001	0.94	<0.001	0.95	<0.001	0.97	<0.001	0.95	<0.001	0.91	<0.001
Policy	Recall	0.80	-	0.66	-	0.60	-	0.59	-	0.74	-	0.74	-	0.60	-	0.53	-
	Call	0.55	0.002	0.35	0.02	0.80	0.001	0.80	0.001	0.73	<0.001	0.74	0.005	0.54	<0.001	0.61	<0.001
	Set upper limit	−0.90	0.002	−0.77	0.005									−0.47	0.014	−0.43	0.025
	No-charge					0.38	0.03	0.41	0.023								
**Regressions**																	
Response variables	Explanatory variables																
Screening rate	Medical/financial resources									0.58	0.003						
	Screening interventions	0.53	0.015	0.60	0.004	0.81	0.001	0.69	0.001					1.14	0.001	1.23	0.005
	Annual household income	0.27	0.017	0.34	0.003			0.24	0.038	0.51	<0.001	0.41	0.002	0.38	<0.001		
	Aging rate					−0.31	0.021			−0.40	0.029			−0.70	<0.001	−0.93	<0.001
Screening interventions	Medical/financial resources	0.62	<0.001	0.67	0.001	0.48	0.016	0.41	0.032	0.48	0.009	0.434	0.014	0.78	<0.001	0.77	0.001
Resource	Aging rate	0.73	<0.001	0.82	<0.001	0.75	<0.001	0.84	<0.001	0.74	<0.001	0.821	<0.001	0.83	<0.001	0.85	<0.001
**Model fit indices**																	
GFI		0.872		0.892		0.943		0.887		0.874		0.876		0.93		0.909	
AGFI		0.738		0.78		0.87		0.789		0.717		0.753		0.85		0.782	
SRMR		0.09		0.091		0.07		0.093		0.111		0.115		0.081		0.084	
CFI		0.955		0.946		1.000		0.937		0.904		0.890		0.945		0.917	
RMSEA		0.087		0.095		<0.001		0.096		0.134		0.144		0.102		0.149	
χ^2^ test (*p* value)		0.128		0.085		0.683		0.077		0.018		0.007		0.060		0.010	

All the values are standardized. Call, sending screening invitations; Recall, re-invitations to the unscreened after the call; Charge-free, providing cancer screenings free of charge; Upper-limit, limiting the number of people who can participate in the cancer screenings. GFI (goodness of fit index); AGFI (adjusted goodness of fit index); SRMR (standardized root mean square of residual); CFI (comparative fit index); RMSEA (root mean square error of approximation).

## Data Availability

Publicly available datasets were analyzed in this study. This data can be found here: [https://www.e-stat.go.jp/en, https://ganjoho.jp/reg_stat/statistics/stat/screening/excel/Pref_Cancer_Screening_Assessment(2015–2019).xlsx] (accessed on 30 June 2022).

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
