# Peer review of "Availability of Financial and Medical Resources for Screening Providers and Its Impact on Cancer Screening Uptake and Intervention Programs"

_ijerph, 2022, doi:10.3390/ijerph191811477_

Round 1
Reviewer 1 Report
Dear Editor,
Thanks for giving me this opportunity to review the manuscript ijerph-1818016 in International Journal of Environmental Research and Public Health. In this manuscript, the author’s presented the “Availability of Financial and Medical Resources for Screening Providers and its Impact on Cancer Screening Uptake and Intervention Programs”.
Recommendation: In my opinion the above mentioned manuscript need to major revision.
My specific comments are as follows:
The manuscript need to edit by native English speaker.
1) Abstract:
What are the study eligibility criteria?
What is study limitation?
2) Introduction
The introduction section was not well written. What is the gap of knowledge?
3) Discussion
Discussion need to compare the result by other studies.
Discuss any limitations of the review processes used.
What is the clinical implication of this study?
Author Response
Reviewer comments:
Reviewer #1:
Abstract:
What are the study eligibility criteria?
Reply:
Thank you very much for pointing out the lack of explanation. We have added a summary of the eligibility criteria starting from line 21 to 25 as follows:
“Data for Japanese municipalities’ medical/financial status, their implementation of screening interventions, and the number of municipality-based cancer screening attendance from April 2016 to March 2017 were obtained from an open database. Five cancer screenings were included: gastric, lung, colorectal, breast, and cervical cancer screening, nationally recommended for population screening in Japan.”
What is study limitation?
Reply:
Thank you very much for pointing out the lack of explanation. We have added a description of the limitations in line 30, as follows:
“One limitation of this study is that it only included screening by municipalities, which may cause selection bias.”
Due to the word limit for the abstract, we added only a part of the limitations in the abstract. We have explained the limitations in detail at the end of the discussion section, in lines 375 to 395.
Introduction:
The introduction section was not well written. What is the gap of knowledge?
Reply:
Thank you very much for your comment and question. Following your suggestions and those of the other reviewers, we have extensively revised our manuscript. We have summarized the knowledge from previous studies in the second paragraph of the introduction and described the gap in the knowledge and its importance in the third paragraph.
The gap in the current knowledge in this instance is how multiple factors related to cancer screening interact to affect screening rates. We think it is necessary to investigate the model of how the factors interact, just as they do in clinical settings, to identify what measures are efficient for improving screening rates.
Discussion:
Discussion need to compare the result by other studies.
Reply:
Thank you very much for your kind advice. In paragraphs 2 - 5 of the discussion, we have added descriptions comparing the results obtained in the present study with previous reports. The added descriptions are as follows:
“Previous studies showed that the number of public health nurses affected the screening rates [15,16]. Our results are consistent with these previous reports, suggesting that the relationship between the number of public health nurses and the screening rates is probably mediated by the screening interventions.” (Lines 319 to 323)
“This result is consistent with previous studies indicating that financial pressure due to the lack of subsidies negatively influences screening rates [15,16].” (Lines 330 to 332)
“This study included the variables related to residents, household income, and age as factors affecting screening rates. Household income positively influenced screening rates for every cancer type, consistent with previous results [3]. The results for age are consistent with previous studies and provide further information. There was an indirect, positive effect through latent variables and a direct negative effect on screening rates, as observed in the model for breast cancer. The sum of these two effects remained positive for screening rates, consistent with previous results [4]. However, the results of this study indicate that age may affect screening rates through multiple mechanisms, suggesting that the effect of age on screening rates may change with the target population and screening system differences.
” (Lines 344 to 353)
“A survey supporting this result was conducted among Japanese municipalities that limited screening participation, with 64% reporting that the limitation was due to the limited capacity of the screening centers and 27% reporting that it was due to financial restrictions [33]. However, the mechanisms at work between these factors have not been analyzed previously.” (Lines 356 to 360)
The results of the current study are consistent with those of previous research and are more informative, regarding the impact of age on screening rates.
Discuss any limitations of the review processes used.
Reply:
Thank you for your kind advice. We have added explanations about the limitations of the review process as follows:
“Finally, there was a limitation in the review process. Other studies may have analyzed the relationship between the screening rates and the factors affecting it using different statistical methods. In addition, since we searched only English and Japanese literature, we did not examine previous studies in other languages. However, to the best of our knowledge, no previous studies have analyzed the entire relationship among factors affecting screening rates as a model, nor are there any studies that used SEM to investigate the relationship between screening rates and the factors affecting them.” (Lines 391 to 398)
What is the clinical implication of this study?
Reply:
Thank you for your question. The clinical implication of this study is that supporting the municipalities’ medical/financial resources may enhance screening interventions and improve cancer screening rates. We have added descriptions to explain this point as follows:
“These results indicate that by improving the financial situation of screening providers and expanding their medical resources, screening providers may strengthen their interventions and thus, improve screening rates. The results of this study suggest that identifying and supporting the financial/medical resources lacking in each municipality could improve screening interventions and, consequently, increase screening rates.” (Lines 404 to 408)

Reviewer 2 Report
The aim of the manuscript was to assess causal relationships between screening rates and factors that can affect them, including unmeasurable ones, such as the availability of medical/financial resources of municipalities or how municipalities provide screening interventions in Japan.
The manuscript is original and well presented. Some minor revisions are requested. In general, the English in the paper can be understood. There are some grammatical and typographic errors, all of which could be corrected during editing. Thus, I consider the manuscript acceptable for publication at this phase with minor revisions.
In particular, the minor revisions regard in the introduction section the lines 40-46, in which authors should specify how these factors influence the screening rates, supporting data with literature studies.
Author Response
Reviewer comments:
Reviewer #2:
Comments and Suggestions for Authors:
The manuscript is original and well presented. Some minor revisions are requested. In general, the English in the paper can be understood. There are some grammatical and typographic errors, all of which could be corrected during editing. Thus, I consider the manuscript acceptable for publication at this phase with minor revisions.
In particular, the minor revisions regard in the introduction section the lines 40-46, in which authors should specify how these factors influence the screening rates, supporting data with literature studies.
Reply:
We appreciate your kind advice. Following your suggestions, we have added descriptions of how the multiple factors affect screening rates. In particular, we added detailed explanations for income and age, supporting data from previous results, which seemed to substantially affect screening rates in the current study. The amended descriptions are as follows:
“For example, higher income is associated with higher cancer screening rates, specifically, the screening rates for cervical and breast cancer in the lowest income quartile were 61.6% and 53.8%, respectively, and in the highest income quartile were 73.4% and 68.3%, respectively [3]. Older age also positively affects screening rates: “men and women 65 years and older had higher rates of any recommended colorectal cancer test (55.8% and 48.5%, respectively) than persons 50 to 64 years (males, 41.0%; females, 31.4%) [4].”” (Lines 44 to 50)